# Molecular Determination of mirRNA-126 rs4636297, Phosphoinositide-3-Kinase Regulatory Subunit 1-Gene Variability rs7713645, rs706713 (Tyr73Tyr), rs3730089 (Met326Ile) and Their Association with Susceptibility to T2D

**DOI:** 10.3390/jpm11090861

**Published:** 2021-08-29

**Authors:** Rashid Mir, Imadeldin Elfaki, Faisel M. Abu Duhier, Maeidh A. Alotaibi, Adel Ibrahim AlAlawy, Jameel Barnawi, Abdullatif Taha Babakr, Mohammad Muzaffar Mir, Hyder Mirghani, Abdullah Hamadi, Pradeep Kumar Dabla

**Affiliations:** 1Department of Medical Lab Technology, Faculty of Applied Medical Sciences, University of Tabuk, Tabuk 71491, Saudi Arabia; jbarnawi@ut.edu.sa (J.B.); a.aldhafri@ut.edu.sa (A.H.); 2Department of Biochemistry, Faculty of Science, University of Tabuk, Tabuk 71491, Saudi Arabia; aalalawy@ut.edu.sa; 3King Faisal Medical Complex Department of Training, Research and Academic Affairs, P.O. Box 2775, Taif 21944, Saudi Arabia; maed96@hotmail.com; 4Department of Medical Biochemistry, Faculty of Medicine, Umm Al-Qura University, Makkah 57039, Saudi Arabia; abdullatiftaha@yahoo.com; 5Department of Basic Medical Sciences, College of Medicine, University of Bisha, Bisha 61992, Saudi Arabia; mirmuzaffar11@gmail.com; 6Internal Medicine and Endocrine, Medical Department, Faculty of Medicine, University of Tabuk, Tabuk 71491, Saudi Arabia; h.mirghani@ut.edu.sa; 7Department of Biochemistry, Govind Ballabh Pant Institute of Postgraduate Medical Education & Research (GIPMER), Associated to Maulana Azad Medical College, Delhi 110002, India; pradeep.dabla@gmail.com

**Keywords:** diabetes mellitus, genome wide association studies, microRNAs, gene variations, miR126 rs4636297, phosphoinositide 3-kinase (PI3K)

## Abstract

Type 2 diabetes is a metabolic disease characterized by elevated blood sugar. It has serious complications and socioeconomic impact. The MicroRNAs are short single-stranded and non-coding RNA molecules. They regulate gene expression at the post-transcriptional levels. They are important for many physiological processes including metabolism, growth, and others. The phosphoinositide 3-kinase (PI3K) is important for insulin signaling and glucose uptake. The genome wide association studies have identified the association of certain loci with diseases including T2D. In this study we have examined the association of miR126 rs4636297 and Phosphoinositide-3-kinase regulatory subunit 1 (PIK3R1) gene Variations rs7713645, rs706713 (Tyr73Tyr), and rs3730089 (Met326Ile) with T2D using the amplification refractory mutation system PCR. Results indicated that there was a significant different (*p*-value < 0.05) in the Mir126 rs4636297 genotypes distribution between cases and controls, and the minor allele of the rs4636297 was also associated with T2D with OR = 0.58, *p*-value < 0.05. In addition results showed that there were significant differences (*p*-value < 0.05) of rs4636297 genotype distribution of patients with normal and patient with abnormal lipid profile. Results also showed that the PIK3R1 rs7713645 and rs3730089 genotype distribution was significantly different between cases and controls with a *p*-values < 0.05. In addition, the minor allele of the rs7713645 and rs3730089 were associated with T2D with OR = 0.58, *p*-value < 0.05. We conclude that the Mir126 rs4636297 and PIK3R1 SNPs (rs7713645 and rs3730089) were associated with T2D. These results need verification in future studies with larger sample sizes and in different populations. Protein-protein interaction and enzyme assay studies are also required to uncover the effect of the SNPs on the PI3K regulatory subunit (PI3KR1) and PI3K catalytic activity.

## 1. Introduction

Diabetes mellitus (DM) is a group of metabolic disorders characterized by increased blood glucose due to insufficient insulin amount or defective insulin action or both [1]. DM is one of the global health issues due to its bad socio-economic effects on the patients, their families and public health [2]. The incidence rate of DM has risen in most countries. It has been estimated in year 2017 that there were more than 400 million individuals suffering from DM [2]. This number is expected to rise to more than 690 million by the year 2045 if no sufficient preventive and control measures were adopted [2]. In term of DM prevalence, the WHO has ranked KSA as number two in the Middle East with seven million individuals suffering from DM and 3 million were reported to be pre-diabetics [3]. The long-term complications of DM include microvascular and macro vascular complications [4]. The microvascular complications include the diabetic nephropathy (DN), diabetic retinopathy (DR) and diabetic neuropathy (DN) [4], whereas the macrovascular complications are the cardiovascular diseases such as coronary artery disease, stroke and peripheral artery disease [4]. Mainly, there are two main types of DM, type 1 DM (T1DM) and type 2 DM (T2DM). T1DM is developed from destruction of pancreatic beta cells by cellular mediated autoimmune process leading to absolute insulin deficiency and represent from 5 to 10% of DM [1,5]. T2DM is induced from defective insulin signaling initiated by insulin resistance in liver, adipose tissues and skeletal muscles. This leads to defective insulin metabolic action and relative insulin deficiency [6]. T2DM is developed by the interaction of environmental and genetic risk factors [7]. Environmental and risk factors of T2DM include aging, obesity, physical inactivity, unhealthy diet [7]. Genome wide association studies (GWAS) have revealed the association of certain loci with the risk to several diseases including DM [8,9,10,11,12,13,14]. MicroRNAs (miRNAs) are short (22 nucleotides length) single-stranded and non-coding RNA molecules. They regulate gene expression at the post-transcriptional levels [15]. MiRNAs regulate physiological process e.g., metabolism, development, proliferation, differentiation and apoptosis [15]. It has been reported that in T2D there was a reduced plasma level of mir126, and there would be loss of endothelial miR-126 [16]. Furthermore, reduced plasma miR-126 has been suggested as a biomarker for T2DM [17]. In addition, the miR-126 rs4636297 SNP was associated with DR in T1DM patients from Brazil [18]. PI3K is a heterodimeric enzyme composed of a catalytic subunit (p110) and a regulatory subunit (p85) [19]. The PI3K regulatory subunit (PIK3R1) comprised a SH3 domain, a breakpoint cluster region homology domain, two SH2 domains, and an inter-SH2 domain [20]. The PI3K is an important in insulin signaling cascade [21], and the PI3K/AKT signaling pathway is important for normal metabolism and its dysregulation results in obesity and T2DM [22]. In the present study we evaluated the association of miR126 (rs4636297) and PIK3R1 gene variations rs7713645, rs706713 (Tyr73Tyr), and rs3730089 (Met326Ile) with T2D.

## 2. Materials and Methods

This study was approved by the research and studies department, directorate of health affairs, Taif, approval No. 229, and by the research ethics committee of the armed forces hospitals, northwestern region, Tabuk, approval No. R & REC2016-115. The study was conducted in the Department of Biochemistry, Faculty of Science in collaboration with Prince Fahd Bin Sultan Research Chair, Department of Medical Lab Technology, Faculty of Applied Medical Sciences, University of Tabuk. All subjects completed the questionnaire as well as informed consent.

### 2.1. Data Collection

This is a case-control study enrolled about 100 subjects each with type 2 diabetes mellitus (T2D) and about 120 normal control subjects for each SNP. T2D was diagnosed on the basis of the WHO criteria. This study included clinically confirmed cases of T2D in Saudi Arabia visiting the armed forces hospital in Tabuk, Al Noor Specialist Hospital in Mecca and the King Faisal Hospitals in Taif for routine checkup. The control subjects were matched healthy volunteers with no history of diabetes or any major clinical disorders and had normal fasting plasma glucose level. The T1D, T2D cases with other significant chronic diseases or malignancies were excluded from the study. The variables that were analyzed from the T2D patients include the case history, age and gender; duration of T2D; glycated hemoglobin (HBA1c); random blood glucose, total cholesterol, Triacylglycerol, high-density lipoprotein-Cholesterol (HDL-C), and low-density lipoprotein cholesterol (LDL-C) concentrations and total cholesterol/HDL-C ratios have been assayed using the standard protocols. The biochemical characteristics of control and cases were shown in Table 1.

### 2.2. Sample Collection and DNA Extraction

From each Subject about 4 mL, a peripheral blood sample was collected in an EDTA tube. Genomic DNA was isolated using the Thermo Scientific Genomic DNA Purification Kit (Waltham, MA, USA) from the whole blood according to the manufacturer’s instructions. The DNA integrity was checked with 0.8 agarose gel electrophoresis and Nanodrop.

### 2.3. Genotyping of SNPs by Amplification-Refractory Mutation System PCR

The microR-126 rs4636297 A > G SNP was genotyped by ARMS-PCR (Figure 1A). The genotyping of three SNP of PIK3R1 gene, the PIK3R1 rs7713645 A→C (Figure 1B), PIK3R1 rs706713 C→T (Figure 1C) and PIK3R1 rs3730089 A→G (Figure 1D) by ARMS-PCR. The primers for all four SNPs were designed by using primer3 software as depicted in Table 2. The ARMS-PCR was done in a reaction volume of 25 µL containing template DNA (50 ng), FO −0. 30 µL, RO −0. 30 µL, RI −0. 20 µL, RI −0. 20 µL of 25 pmol of each primers and 10 µL from GoTaq^®^ Green Master Mix (cat no M7122) (Promega, Madison, WI, USA). The final volume of 25 µL was adjusted by adding nuclease free ddH_2_O. Finally, the 2 µL of DNA was added from each patient. The thermocycling conditions used were at 95 °C for 10 min followed by 40 cycles of 95 °C for 35 s, annealing temperature PIK3R1 rs706713 C→T (58 °C) PIK3R1 rs3730089 A→G (60 °C), PIK3R1 rs7713645 A→C (55 °C) and microR-126 rs4636297 A > G (58 °C) for 40 s, 72 °C for 43 s followed by the final extension at 72 °C for 10 min. PCR products were separated on 2% agarose gel stained with 2 µL of sybre safe stain and visualized on a UV transilluminator from Biorad (Hercules, CA, USA).

## 3. Statistical Analysis

T2D patients and controls were compared by statistical analysis using the SPSS 16.0 software package. Chi-square analysis and Fisher exact test were carried out to compare microR-126 rs4636297 A > G, PIK3R1 rs706713 C→T, PIK3R1 rs3730089 A→G, PIK3R1 rs7713645 A→C gene polymorphism frequency with several clinical aspects. The Hardy-Weinberg equilibrium was tested by a χ^2^ test to compare the observed genotype frequencies within the case-control groups. *p*-value was considered to be significant when it was <0.05.

## 4. Results

Results indicated that there was a significant difference in Mir126 rs4636297 genotypes (AA, AG, GG) distribution between cases (65, 43, 05) and controls (47, 54, 13) with *p*-value = 0.021 (Table 3). In the co-dominant model, it was shown that the GG genotype was associated with T2D with an OR (95% CI) = O.27 (0.09 to 0.83), RR = 0.58(0.40 to 0.83), *p*-value = 0.02 (Table 4). In the dominant model, (GA + GG) was associated with T2D with an OR (95% CI) = 0.51 (0.30–0.87), RR= 0.72 (0.55–0.94), *p*-value = 0.014 (Table 4). The G allele was also associated with T2D with OR (95% CI) = 0.58 (0.38–0.87), RR = 0.77 (0.64–0.93), *p*-value = 0.009 (Table 4). In addition, results showed that there were significant differences (*p*-values < 0.05) in the lipid profile in patients with normal and patient with abnormal lipid profile (Table 5).

Results showed that there was a significant difference in PI3KR1 rs7713645 genotype (AA, AC, CC) distribution between cases (08, 80, 12) and controls (37, 60, 11) with a *p*-value = 0.0001 (Table 6). Results also showed that in the Co-dominant model, the CA genotype is associated with T2D with an OR = 6.16 (2.67 to 14.20), RR = 1.91(1.51 to 2.42), *p*-value = 0.0001 (Table 7). In the dominant model, the (CA + CC) was associated with T2D with an OR = 5.99 (2.62–13.66), RR = 1.88(1.51–2.35), *p*-value = 0.0001 (Table 6). Moreover, the C allele was associated with T2D with OR = 1.77 (1.19–2.61), RR = 1.32 (1.08–1.60), *p*-value = 0.004 (Table 7). Moreover, our result showed that there were significant differences (*p*-value < 0.05) in the lipid profile in patients with normal and patient with abnormal lipid profile (Table 8). Results indicated there was a significant difference in the genotype distribution between patients with age over 20 and less than 40 years old.

Results showed that there was a significant differences in the rs706713 SNP genotype (CC, CT, TT) distribution between the cases (68, 33, 0), and controls (53, 45, 03) with *p*-value = 0.03 (Table 9). Results showed that in the dominant model, the (TC + TT) was associated with OR (95% CI) = 0.53 (0.3–0.94), RR = 0.73 (0.56–0.96), *p*-value = 0.032 (Table 10). Furthermore, the T allele was associated with T2D with OR (95% CI) = 0.36 (0.22–0.58), RR = 0.59 (0.47–0.73), *p*-value = 0.0001 (Table 10). Results indicated there was a significant difference in the genotype distribution between patients with aged over 20 and patients aged less than 40 years old (Table 11).

Results indicated that there was a significant differences in the rs3730089 SNP genotype (GG, GA, AA) between cases (09, 49, 42) and controls (18, 69, 35) with a *p*-value = 0.03 (Table 12). The A allele was associated with T2D with OR (95% CI) = 1.49 (1.01–2.21), RR = 1.19 (1.01–1.41), *p*-value = 0.04 (Table 13). We could not see significant differences in genotype distribution of cases with different parameters (Table 14).

## 5. Discussion

DM is one of the important causes of morbidity and mortality all over the world. It has serious impact on public health system as well as socioeconomic status of patients and their families. The incidence and prevalence rates of DM in KSA were increased [23]. MiRNAs are short, noncoding, single stranded RNA molecules that have been involved in crucial biological process such as development, metabolism, differentiation and apoptosis [15]. Gene variations of MiRNAs were associated with many diseases including T2D [11,24]. The MiR-126 gene is found within the gene of the epidermal growth factor-like protein 7 (EGFL7) in chromosome 9q34.3 [25]. This polymorphism has been reported to be associated with ischemic stroke in Chinese population [25]. In addition, reduced mir126 plasma levels have been suggested as a biomarker for early identification of individual susceptibility to T2D [26]. Our results indicated that Mir126 rs4636297 genotypes distribution was significantly different (*p*-value < 0.05) between T2D cases and the control (Table 3). The GG genotype of the rs4636297 was associated with T2D (Table 4). The G of the rs4636297 allele was also associated with T2D (Table 4). This result maybe in partial agreement with previous studies suggested that the miR-126 rs4636297 SNP was associated with diabetic retinopathy in different populations [27,28]. In addition, results showed that there were significant differences (*p*-value < 0.05) in the lipid profile in cases with normal and cases with abnormal lipid profile (Table 5). This result may also be consistent with studies suggested that miR-126 is involved lipid metabolism [29,30] because there was an association between miR-126 and LDL-C circulating levels [30]. This rs4636297 SNP may reduce the amount of miR-126 in plasma; the reduced miR-126 was described as an early indication of individual risk to T2D [26]. However, the effect of this SNP on miR-126 remains to be investigated in future studies.

PI3-kinases are family of lipid kinases that catalyze the addition of phosphate to the 3-OH position of cellular membrane inositol lipids [31]. PI3-kinases are important for several physiological processes such as metabolism, proliferation, survival and growth [31,32]. The PI3-kinases are heterodimer enzymes composed of regulatory subunit (e.g., p85α, p85β, p55α, p50α, or p55γ) and catalytic subunits (e.g., p110α, β or δ) [33]. There are several isoforms of the PI3-kinases regulatory and catalytic subunits [31,33]. This enzyme is very important for insulin metabolic action [31]. It has been reported that construction of dysfunctional PI3-kinases or pharmacological inhibition of PI3-kinases resulted in dysregulation of glucose transport, synthesis of lipid and glycogenesis [31]. Furthermore, abolishment of PI3-kinase stimulation by insulin in animal model resulted in increased body fat and T2D development [31], whereas deletion of regulatory subunit of the PI3-kinases in muscles resulted in impairment of insulin signaling, deceased uptake of glucose and increased glucose plasma levels [19,34]. Our results indicated that there was a significant difference (*p*-value < 0.05) in the SNP rs7713645 (in PIKR intron) genotype distribution between cases and controls (Table 6), and that the CA and CC genotypes and the C allele of the rs7713645 were associated with T2D (Table 7). Results also showed that there is a significant difference (*p*-value < 0.05) in SNP rs7713645 and the genotype distributions between cases with normal and cases with abnormal lipid profile (Table 8), and there was a significant differences in rs7713645 genotype distribution between cases with age > 20 and cases with age > 40 years (Table 8). This result is consistent with previous studies reported that the SNP rs7713645 is associated with higher MBI, increased body fat, and higher fasting glucose levels [35]. This result is also in agreement with a study that reported an association of rs7713645 with T2D in Turkish population [36].

Results indicated that there was a significant difference (*p*-value < 0.05) in the rs706713 (in PIKR exon1) genotype distribution between cases and controls (Table 9). The rs706713 CT genotype and the T allele were associated with T2D (Table 10). There was also a significant difference between cases with age > 20 and cases with age > 40 years (Table 11). The SNP rs706713 is a silent mutation, i.e., there is no change in the amino acid (Tyr73Tyr) [36]. We do not have explanation why there was a significant difference in rs706713 genotype distribution between cases and controls, and a significant difference between cases with age > 20 and cases with age > 40 years old. However, this result is consistent with Karadoğan et al. who indicated association of rs706713 with T2D in Turkish population [36].

Results also showed that the exon SNP rs3730089 (Met326Ile, in exon 6) genotype distribution were significantly different between cases and control (Table 12) and that the A allele of the rs3730089 was associated with T2D (Table 13). A previous study [37] examined the expression of PI3-kinase regulatory subunit (p85α) and protein-protein interaction of the insulin receptor substrate (IRS-1) with p85α using the yeast two hybrid assay [37,38]. It was reported that the SNP rs3730089 (Met326Ile) resulted in reduced p85α expression but increased binding between p85α and IRS1 [37], and therefore the rs3730089 may have minor impact on insulin signaling and glucose uptake [37]. This result is consistent with previous studies showed potential association of rs3730089 SNP with T2D [36] and insulin signaling [37]. However, another study reported no association of rs3730089 SNP with T2D in north Indian population [39]. This difference may be due to different sample sizes or different populations. We did no find significant differences in rs3730089 genotype distribution of cases with different parameters (Table 14). This is probably due to the small sample size used in this study.

Limitations of this cross-sectional study include the small sample size and the blood samples were collected from patients one time and maybe after the blood biochemistry was already controlled.

Future longitudinal studies with larger sample sizes and in different populations to examine the effects of both the miR-126 SNP and one or more PIKR risk alleles on susceptibility to T2D, T2D complications and lipid metabolism are strongly recommended. In addition, further-protein protein interactions studies using the techniques such as yeast-two hybrid, phage display, x-ray crystallography, protein NMR spectroscopy [38,40,41,42,43] are required to examine the effect of SNPs in PI3KR1 gene on the PI3K structure, function and interaction with other proteins.

In summary, our study indicated that the Mir126 rs4636297 was associated with T2D. Results also showed that there was a significant difference in rs4636297 in genotype distribution between cases with normal and cases with abnormal lipid profile. Results also showed that the PI3KR1 SNPs rs7713645, rs3730089 were associated with T2D. The PI3KR1 SNP rs7713645 may also be associated with abnormal lipid profile. These results require further verification in future studies with larger sample sizes and in different populations.

## Figures and Tables

**Figure 1 jpm-11-00861-f001:**
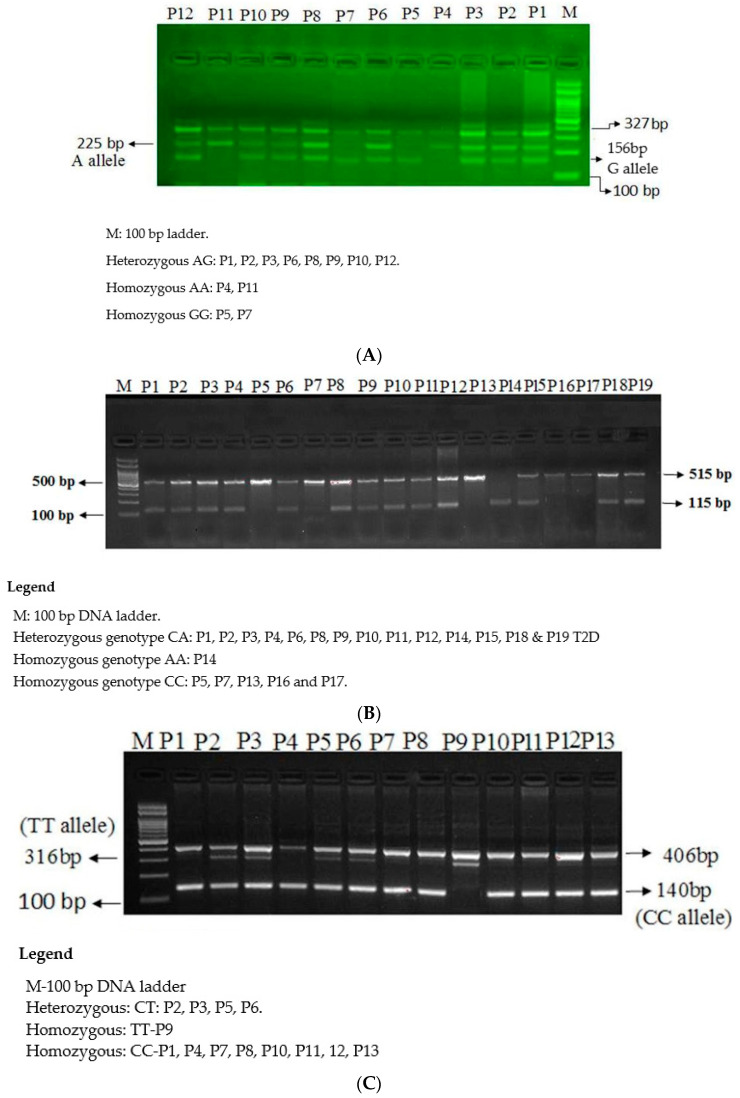
(**A**) Genotyping of the Mir126 rs4636297 using the ARMS-PCR, (**B**) Genotyping of the PIK3R1 rs7713645 using the ARMS-PCR, (**C**) Genotyping of the PIK3R1 rs706713 (Tyr73Tyr) using the ARMS-PCR, (**D**) Genotyping of the PIK3R1 rs3730089 (Met326Ile) using the ARMS-PCR.

**Table 1 jpm-11-00861-t001:** The glycated hemoglobin (HBA1c), Triglyceride (TG), cholesterol (choles.), low density lipoprotein cholesterol (LDL-C), high density lipoprotein cholesterol (HDL-C), fasting blood sugar (FBS), random blood sugar (RBS) and vitamin D of healthy controls and T2D patients.

	HBA1c %	TG mg/dL	Choles. mg/dL	LDL-C mg/dL	HDL-C mg/dL	FBS mg/dL	
Controls	5	135	153	74	57.0	89	
	HBA1c %	TG mg/dL	Choles mg/dL	LDL-C mg/dL	HDL-C mg/dL	RBS mg/dL	Vit. D ng/mL
Cases	9	178	198	130	44	224	28

**Table 2 jpm-11-00861-t002:** Primer sequence of microR-126 SNP and PIK3R1 gene SNPs.

Primer Sequence of microR-126 rs4636297 A > G SNP			
mi126Fo	5-GGATAGGTGGGTTCCCGAGAACTG-3	327 bp	58 °C
mi126Ro	5-TCTCAGGGCTATGCCGCCTAAGT-3		
mi126FI-G	5-TTCAAACTCGTACCGTGAGTAATAATGAGC-3	156 bp	
mi126RI-A	5-GTTTTCGATGCGGTGCCGTGGAAGA-3	225 bp	
Primer Sequence of PIK3R rs7713645 A > C SNP			
PIK3R1-F1	5-CCTACACCAACCCCATTCAGC-3	518 bp	58 °C
PIK3R1-A	5-ACACTCAAATGCTGAATGTGAAAAGTT-3		
PIK3R-F2	5-GGTTTCCCAAGGCATGTTATTGTCAC-3	115 bp	
PIK3R-R2C	5-TAGTCAATGTTTGTGATTTATTGCAGCC-3		
Primer Sequence of PIK3R rs706713 C > T SNP			
PIK3R1-Fo	5-TAAAAACGTAAAATCAGACTGCTCTG-3	406bp	58 °C
PIK3R1-Ro	5-TGACCTTGTTGTTCAACATCTGC-3		
PIK3R1C-FI	5-GGGACTTTCCGGGAACTTAC-3	140bp	
PIK3R1T-RI	5-GAGATTTTTTTCCTTCCAATATATTCTACA-3	316bp	
Primer Sequence of PI3KR1 rs3730089 G > A SNP			
PIK3R-F1	CATGGCCAGCCCAATTTATTTGTTC	490 bp	60 °C
PIK3R-R	CGTCTTTGGAAGAGAACCAACTATG		
PIK3R-F1A	GCCAACAACGGTATGAATAACAATA	200 bp	
PIK3RI-C	GTACCATTCAGCATCTTGTAAGGAC	342 bp	

**Table 3 jpm-11-00861-t003:** Association of MicroRNA-126 rs4636297 A > G genotypes between T2D patients and healthy controls.

Subjects	*n*	AA	GA	GG	A	G	χ^2^	df	*p*-Value
T2D patients	113	65 (57.52%)	43 (38%)	05 (4.42%)	0.77	0.23	7.69	2	**0.021**
Controls	114	47 (41.22%)	54 (47.36%)	13 (11.40%)	0.65	0.35			

Statistically significant *p*-values (*p* < 0.05) were indicated with bold numbers.

**Table 4 jpm-11-00861-t004:** Association of MicroRNA-126 rs4636297 A > G gene variation with T2D.

Mode of Inheritance	Controls (*n* = 114)	Cases (*n* = 113)	OR (95% CI)	RR (95% CI)	*p*-Value
Co-dominant model					
MiR-AA	47 (41.22%)	65 (57.52%)	1 (ref.)	1 (ref.)	
MiR-GA	54 (47.36%)	43(38%)	0.57 (0.33 to 0.99)	0.76 (0.57 to 1.00)	0.05
MiR-GG	13 (11.40%)	05 (4.42%)	0.27 (0.09 to 0.83)	0.58 (0.40 to 0.83)	**0.02**
Dominant model					
MiR-AA	47 (41.22%)	65 (57.52%)	1 (ref.)	1 (ref.)	
MiR (GA + GG)	67 (58.77%)	48 (42.47%)	0.51 (0.30–0.87)	0.72 (0.55–0.94)	**0.014**
Recessive model					
MiR (AA+ GA)	101 (88.59%)	108 (95.57%)	1 (ref.)	1 (ref.)	
MiR-GG	13 (11.40%)	05 (4.42%)	0.35 (0.12–1.04)	0.69 (0.48–0.92)	0.067
Allele					
MiR-A	148	173	1 (ref.)	1 (ref.)	
MiR-G	78	53	0.58 (0.38–0.87)	0.77 (0.64–0.93)	**0.009**

Abbreviations: OR = Odds Ratio, RR = Risk Ratio, CI = Confidence interval; Statistically significant *p*-values (*p* < 0.05) were indicated with bold numbers.

**Table 5 jpm-11-00861-t005:** Clinical associations of MicroRNA rs4636297 G > A genotypes with clinic variables features of the T2D patients.

Subjects	*n* = 113	AA	GA	GG	χ^2^	df	*p*-Value
Association with gender							
Males	80	50	26	04	3.6	2	0.160
Females	33	15	17	01			
Association with Age							
Age > 20	27	15	10	02	0.75	2	0.068
Age > 40	86	50	33	03			
Association with RBS mg/dL							
RBS < 140	34	22	10	02	1.62	2	0.444
RBS > 140	79	43	33	03			
Association with Cholesterol mg/dL							
Cholesterol < 200	81	50	30	01	7.54	2	**0.023**
Cholesterol > 200	32	15	13	04			
Association with HDL-C mg/dL							
HDL-C < 55	79	48	30	01	6.4	2	**0.048**
HDL-C > 55	34	17	13	04			
Association with LDL-C mg/dL							
LDL < 100	23	06	15	04	26.1	2	**0.0001**
LDL > 100	77	59	28	01			
Association with TG mg/dL							
TG < 200	61	47	13	01	20.87	2	**0.0001**
TG > 200	52	18	30	04			
Association with HBA1c %							
HBA1c < 6	27	15	10	2	0.25	2	0.882
HBA1c > 6	86	50	33	3			
Association with Vitamin D ng/mL							
Vit.D < 30	18	03	13	02	2.54	2	0.28
Vit.D > 30	14	1	13	0			

Statistically significant *p*-values (*p* < 0.05) were indicated with bold numbers.

**Table 6 jpm-11-00861-t006:** Association of PIKR rs7713645 A > C genotypes between T2D patients and healthy controls.

Subjects	*n*	AA	CA	CC	A	C	χ^2^	df	*p*-Value
PIKR patients	100	8 (8%)	80 (80%)	12 (12%)	0.48	0.52	21.31	2	**0.0001**
Controls	108	37 (34.25%)	60 (55.55%)	11 (10.18%)	0.62	0.38			

Statistically significant *p*-values (*p* < 0.05) were indicated with bold numbers.

**Table 7 jpm-11-00861-t007:** Association of PI3KR1 rs7713645 A > C gene variation with T2D.

Mode of Inheritance	Controls (*n* = 108)	Cases (*n* = 100)	OR (95% CI)	RR (95% CI)	*p*-Value
Co-dominant model					
PIKR-AA	37	8	1 (ref.)	1 (ref.)	
PIKR-CA	60	80	6.16 (2.67 to 14.20)	1.91 (1.51 to 2.42)	**0.0001**
PIKR-CC	11	12	5.04 (1.64 to 15.45)	1.71 (1.09 to 2.69)	**0.0046**
Dominant model					
PIKR-AA	37	8	1 (ref.)	1 (ref.)	
PIKR-(CA + CC)	71	92	5.99 (2.62–13.66)	1.88 (1.51–2.35)	**0.0001**
Recessive model					
PIKR-(AA + CA)	97	88	1 (ref.)	1 (ref.)	
PIKR-CC	11	12	1.20 (0.50–2.86)	1.09 (0.70–1.71)	0.67
Allele					
PIKR-A	134	96	1 (ref.)	1 (ref.)	
PIKR-C	82	104	1.77 (1.19–2.61)	1.32 (1.08–1.60)	**0.004**

Abbreviations: OR = Odds Ratio, RR = Risk Ratio, CI = Confidence interval; Statistically significant *p*-values (*p* < 0.05) were indicated with bold numbers.

**Table 8 jpm-11-00861-t008:** Clinical associations of PI3KR1 rs7713645 A > C SNP genotypes with clinic variables features of the T2D patients.

Subjects	*n* = 100	AA	CA	CC	χ^2^	df	*p*-Value
Association with gender		8	80	12			
Males	59	4	47	8	0.56	2	0.755
Females	41	4	33	4			
Association with Age		8	80	12			
Age > 20	14	04	08	02	9.75	2	**0.0076**
Age > 40	86	04	72	10			
Association with RBS mg/dL		8	80	12			
RBS < 140	34	05	27	02	4.5	2	0.102
RBS > 140	66	03	53	10			
Association with Cholesterol mg/dL		8	80	12			
Cholesterol < 200	64	05	60	02	16.14	2	**0.0003**
Cholesterol > 200	36	3	20	10			
Association with HDL-C mg/dL		8	80	12			
HDL < 55 mg	72	05	63	04	11.7	2	**0.0039**
HDL > 55 mg	28	03	17	08			
Association with LDL-C mg/dL							
LDL-C < 100	23	05	14	04	9.14	2	**0.010**
LDL-C > 100	77	03	66	08			
Association with TG mg/dL							
TG < 200	73	03	62	08	6.31	2	**0.045**
TG > 200	27	05	18	04			
Association with HBA1c %							
HBA1c < 6	01	0	1	0	0.25	2	0.882
HBA1c > 6	99	08	79	12			
Association with Vitamin D ng/mL							
Vit.D < 30	18	03	13	02	2.54	2	0.28
Vit.D > 30	14	1	13	0			

Statistically significant *p*-values (*p* < 0.05) were indicated with bold numbers.

**Table 9 jpm-11-00861-t009:** Genotype frequency of PI3KR1 SNP rs706713 T > C polymorphism of study cohorts.

Variables/Genotype	C/C	T/C	T/T	χ^2^	df	*p*-Value
PIKR patients	68 (67.3%)	33 (32.7%)	0 (0%)	6.71	2	**0.03**
Controls	53 (52.5%)	45 (44.6%)	3 (2.9%)			

Statistically significant *p*-values (*p* < 0.05) were indicated with bold numbers.

**Table 10 jpm-11-00861-t010:** Association of PI3KR1 SNP rs706713 C > T gene variation with PIKR.

Mode of Inheritance	Cases (*n* = 101)	Controls (*n* = 101)	OR (95% CI)	RR (95% CI)	*p*-Value
Co-dominant					
PIKR-CC	68	53	1 (ref.)	1 (ref.)	
PIKR-TC	33	45	0.57 (0.32–1.01)	0.75 (0.57–1.0)	0.056
PIKR-TT	00	03	0.11 (0.005–2.2)	0.43 (0.35–0.53)	0.149
Dominant					
PIKR-CC	68	53	1 (ref.)	1 (ref.)	
PIKR-(TC + TT)	33	48	0.53 (0.3–0.94)	0.73 (0.56–0.96)	**0.032**
Recessive					
PIKR-(CC + TC)	101	98	1 (ref.)	1 (ref.)	
PIKR-TT	00	03	0.13 (0.007–2.7)	0.49 (0.42–0.56)	0.193
Allele					
PIKR-C	269	151	1 (ref.)	1 (ref.)	
PIKR-T	33	51	0.36 (0.22–0.58)	0.59 (0.47–0.73)	**0.0001**

Abbreviations: OR = Odds Ratio, RR = Risk Ratio, CI = Confidence interval; Statistically significant *p*-values (*p* < 0.05) were indicated with bold numbers.

**Table 11 jpm-11-00861-t011:** Associations of covariates with PI3KR1 SNP rs706713 genotypes.

Subjects	*n* = 101	C/C	T/C	T/T	χ^2^	df	*p*-Value
Association with gender							
Males	69	42	27	0	4.13	2	0.126
Females	32	26	06	0			
Association with Age							
Age > 20	17	08	09	0	8.47	2	**0.014**
Age > 40	84	60	14	0			
Association with RBS mg/dL							
RBS < 140	27	14	13	0	2.13	2	0.344
RBS > 140	51	35	16	0			
Association with Cholesterol mg/dL							
Cholesterol < 200	53	32	21	0	0.02	2	0.990
Cholesterol > 200	29	18	11	0			
Association with HDL-C mg/dL							
HDL < 55	69	44	25	0	0.84	2	0.657
HDL > 55	13	10	03	0			
Association with LDL-C mg/dL							
LDL < 100	31	21	10	0	0.0	2	1
LDL > 100	52	35	17	0			
Association with TG mg/dL							
TG < 200 mg	63	43	20	0	0.17	2	0.918
TG > 200 mg	19	12	07	0			
Association with HBA1c %							
HBA1c < 6 mg	02	1	1	0	0.28	2	0.869
HBA1c > 6 mg	99	67	32	0			
Association with Vitamin D ng/mL							
Vit.D < 30	15	11	04	0	0.6	2	0.740
Vit.D > 30	15	09	06	0			

Statistically significant *p*-values (*p* < 0.05) were indicated with bold numbers.

**Table 12 jpm-11-00861-t012:** Genotype frequency of PI3KR1 SNP rs3730089 G > A polymorphism of study cohorts.

Variables/Genotype	G/G	G/A	A/A	χ^2^	df	*p*-Value
PIKR patients	9 (9%)	49 (49%)	42 (42%)	6.71	2	**0.03**
Controls	18 (14.8%)	69 (56.5%)	35 (28.7%)			

Statistically significant *p*-values (*p* < 0.05) were indicated with bold numbers.

**Table 13 jpm-11-00861-t013:** Association of PI3KR1 SNP rs3730089 G > A gene variation with PIKR.

Mode of Inheritance	Cases (*n* = 100)	Controls (*n* = 101)	OR (95% CI)	RR (95% CI)	*p*-Value
Co-dominant					
PIKR-GG	9	18	1 (ref.)	1 (ref.)	
PIKR-GA	49	69	1.42 (0.58–3.42)	1.14 (0.83–1.54)	0.434
PIKR-AA	42	35	2.4(0.95–6.0)	1.46 (1.02–2.1)	0.061
Dominant					
PIKR-GG	9	18	1 (ref.)	1 (ref.)	
PIKR-(GA + AA)	91	104	1.75 (0.74–4.08)	1.25 (0.92–1.68)	0.196
Recessive					
PIKR-(GG + GA)	58	87	1 (ref.)	1 (ref.)	
PIKR-AA	42	35	1.8 (1.03–3.14)	1.32 (0.99–1.74)	**0.039**
Allele					
PIKR-G	67	105	1 (ref.)	1 (ref.)	
PIKR-A	133	139	1.49 (1.01–2.21)	1.19 (1.01–1.41)	**0.040**

Abbreviations: OR = Odds Ratio, RR = Risk Ratio, CI = Confidence interval; Statistically significant *p*-values (*p* < 0.05) were indicated with bold numbers.

**Table 14 jpm-11-00861-t014:** Associations of covariates with PI3KR1 SNP rs3730089 genotypes.

Subjects	*n* = 100	G/G	G/A	A/A	χ^2^	df	*p*-Value
Association with gender							
Males	62	3	30	29	4.04	2	0.132
Females	38	6	19	13			
Association with Age							
Age > 20	15	0	6	9	3.24	2	0.197
Age > 40	85	9	43	33			
Association with RBS mg/dL							
RBS < 140 mg	25	3	8	14	2.25	2	0.324
RBS > 140 mg	54	5	27	22			
Association with Cholesterol mg/dL							
Cholesterol < 200	50	5	21	24	0.82	2	0.663
Cholesterol > 200	31	2	16	13			
Association with HDL-C mg/dL							
HDL < 55	70	7	32	31	1.32	2	0.516
HDL > 55	11	0	5	6			
Association with LDL-C mg/dL							
LDL < 100	27	4	11	12	2.02	2	0.364
LDL > 100	54	3	26	25			
Association with TG mg/dL							
TG < 200 mg	61	6	24	31	4.01	2	0.134
TG > 200 mg	20	1	13	6			
Association with HBA1c %							
HBA1c < 6 mg	1	0	1	0	1.07	2	0.585
HBA1c > 6 mg	98	9	47	42			
Association with Vitamin D ng/mL							
Vit.D < 30	14	2	7	5	0.94	2	0.625
Vit.D > 30	17	2	6	9			

## Data Availability

Not applicable.

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
