# Peer review of "Molecular Determination of mirRNA-126 rs4636297, Phosphoinositide-3-Kinase Regulatory Subunit 1-Gene Variability rs7713645, rs706713 (Tyr73Tyr), rs3730089 (Met326Ile) and Their Association with Susceptibility to T2D"

_jpm, 2021, doi:10.3390/jpm11090861_

Round 1

Reviewer 1 Report

Mir and coworkers have investigated the association of genetic variants with the risk of type 2 diabetes. With increased access to genetic information, along with a reduction in costs for whole genome sequencing, the role of single nucleotide polymorphisms in disease will become increasingly relevant. We currently have only a poor understanding of why some people without risk factors become diabetics, and why some people with multiple risk factors do not. We have long suspected that genetic variation must explain this, and this study adds to that body of knowledge.

Major points:

This study was done very well, and used a highly reliable technique (ARMS-PCR) to identify single nucleotide polymorphisms in a miRNA and in PIKR. The statistical analysis is appropriate and the results are very interesting and important.

I can imagine that some of the members of the group had more than one risk allele. Did you look at the effect of this, and the potentially additive effects? It would be interesting to see if people with both the miRNA variant AND one or more PIKR risk alleles were more severely affected, had an earlier onset of disease, had greater effects on lipid homeostasis etc.

The discussion was very good and gave plausible explanations for the association of the risk alleles and T2DM.

Minor points:

  • Some of the blots have red lines on them. Is there any significance to this? Please mention this in your manuscript or redo the electrophoresis.
  • You were very careful to use “association” and “correlation” in your manuscript. This is very important.
  • You should put a space between the word Table and the Table numbers; e.g. Table5 should be Table 5.
  • You have asterisks on statistically significant associations, but it would be helpful to make these bold so they are easy to spot.
  • The English is very good, but would benefit from proof reading by a native speaker.

Author Response

Dear Editor,

Thank you very much for considering our manuscript in the journal of personalized medicine. The comments are very helpful and we tried our level best answer each and every raised by the reviewers.

Please find below systematic responses and answers to reviewer's comments.

Best regards

Comments and Suggestions for Authors

Dear reviewer 1,

Thank you very much for the comments that are very helpful and greatly improved our manuscript.

Please find below systematic responses and answers to reviewer's comments.

Best regards

Mir and coworkers have investigated the association of genetic variants with the risk of type 2 diabetes. With increased access to genetic information, along with a reduction in costs for whole genome sequencing, the role of single nucleotide polymorphisms in disease will become increasingly relevant. We currently have only a poor understanding of why some people without risk factors become diabetics, and why some people with multiple risk factors do not. We have long suspected that genetic variation must explain this, and this study adds to that body of knowledge.

Reviewer1  

Major points:

This study was done very well, and used a highly reliable technique (ARMS-PCR) to identify single nucleotide polymorphisms in a miRNA and in PIKR. The statistical analysis is appropriate and the results are very interesting and important.

 REVIEWER 1  

I can imagine that some of the members of the group had more than one risk allele. Did you look at the effect of this, and the potentially additive effects? It would be interesting to see if people with both the miRNA variant AND one or more PIKR risk alleles were more severely affected, had an earlier onset of disease, had greater effects on lipid homeostasis etc.

Authors

Thank you very much for the valuable comments. Yes, sure this would be a very good idea to examine the effect of both SNPs in miR126 gene and in PIKR gene as this might increase the susceptibility to T2D. This is because the miR126 plasma concentration has been suggested as a biomarker for T2D and that the miR-126 rs4636297 SNP was associated with diabetic retinopathy in different populations. Moreover, PIK is important for insulin metabolism and insulin signaling. Unfortunately, we did not conduct this study, but we will consider it in our future projects and we added it to the recommendation of the study. Changes are highlighted in green.    

 REVIEWER-1

The discussion was very good and gave plausible explanations for the association of the risk alleles and T2DM.

 Minor points:

 Reviewer1

  • Some of the blots have red lines on them. Is there any significance to this? Please mention this in your manuscript or redo the electrophoresis.

Authors

Thank you very much, the figures are corrected. 

Reviewer 1

  • You were very careful to use “association” and “correlation” in your manuscript. This is very important.
  • You should put a space between the word Table and the Table numbers; e.g. Table5 should be Table 5.

Authors

Thank you very much, done

Reviewer 1

You have asterisks on statistically significant associations, but it would be helpful to make these bold so they are easy to spot.

Authors

Thank you very much. Done.

  • The English is very good, but would benefit from proof reading by a native speaker.

Authors

A colleague from our department has revised the English. Authors

Thank you very much

Reviewer 2

Thank you very much for the comments that are very helpful and greatly improved our manuscript.

Please find below systematic responses and answers to reviewer's comments.

Best regards

Reviewer 2

Major comments.

Page 3, lines 87-88.
The previous study showed that the PIK3R1 SNPs rs706713, rs3730089, and rs7713645 were significantly associated with T2D in the Turkish population (Karadogan, A.H., et al., Adv Clin Exp Med, 2018;27:921-927). The authors are recommended to describe which racial groups were studied to replicate the previous study findings, highlighting what they have newly discovered in the present study.

Authors

Thank you very much. The study included T2D patients visiting the armed forces hospital in Tabuk, Al Noor Specialist Hospital Mecca, and the King Faisal Hospitals in Taif for routine checkup. They were all from Saudi population from different Saudi tribes. Our results indicated that the Mir126 rs4636297 and the PI3KR1 SNPs rs7713645, rs3730089 were associated with T2D in Saudi population. Results also showed that there was a significant difference in rs4636297 and rs7713645 genotype distributions between cases with normal and cases with abnormal lipid profile in Saudi population.

Authors

Thank you very much. Done  

Reviewer 2 Report

The authors showed that the mirRNA-126 rs4636297 was associated with T2D and that there was a significant difference in rs4636297 in genotype distribution between cases with normal and cases with abnormal lipid profile. They also showed that the PIK3R1 SNPs rs7713645, rs3730089 were associated with T2D and that the SNP rs7713645 may also be associated with abnormal lipid profile.

Major comments.

Page 3, lines 87-88.
The previous study showed that the PIK3R1 SNPs rs706713, rs3730089, and rs7713645 were significantly associated with T2D in the Turkish population (Karadogan, A.H., et al., Adv Clin Exp Med, 2018;27:921-927). The authors are recommended to describe which racial groups were studied to replicate the previous study findings, highlighting what they have newly discovered in the present study.

Page 3, lines 85-86.
The clinical and biochemical characteristics of the T2D cases and the control studied should be shown in a new Table.

Page 6, lines 147-148.
Page 7, lines 149-150. 
It is recommended that the authors adjust for BMI when they assess whether variants of mirRNA-126 and PIK3R1 are associated with T2D and metabolic traits.

Minor comments.

Page 16, line 282.
The authors should replace the description “table 9” with “table 8”.

Page 2, line 46.
The expression "important global health concerns" does not take an indefinite article.

Page 17, line 307.
The expression "PI3K SNPs" should be replaced with "PIK3R1 SNPs".

Author Response

Dear Editor,

Thank you very much for considering our manuscript in the journal of personalized medicine. The comments are very helpful and we tried our level best answer each and every raised by the reviewers.

Please find below systematic responses and answers to reviewer's comments.

Best regards

Comments and Suggestions for Authors

Reviewer 2

Thank you very much for the comments that are very helpful and greatly improved our manuscript.

Please find below systematic responses and answers to reviewer's comments.

Best regards

Reviewer 2

Major comments.

Page 3, lines 87-88.
The previous study showed that the PIK3R1 SNPs rs706713, rs3730089, and rs7713645 were significantly associated with T2D in the Turkish population (Karadogan, A.H., et al., Adv Clin Exp Med, 2018;27:921-927). The authors are recommended to describe which racial groups were studied to replicate the previous study findings, highlighting what they have newly discovered in the present study.

Authors

Thank you very much. The study included T2D patients visiting the armed forces hospital in Tabuk, Al Noor Specialist Hospital Mecca, and the King Faisal Hospitals in Taif for routine checkup. They were all from Saudi population from different Saudi tribes. Our results indicated that the Mir126 rs4636297 and the PI3KR1 SNPs rs7713645, rs3730089 were associated with T2D in Saudi population. Results also showed that there was a significant difference in rs4636297 and rs7713645 genotype distributions between cases with normal and cases with abnormal lipid profile in Saudi population.

Reviewer 2

Page 3, lines 85-86.
The clinical and biochemical characteristics of the T2D cases and the control studied should be shown in a new Table.

Authors

A table showing the clinical and biochemical characteristics of the T2D cases and the control is added to the study (Table1).

Reviewer 2

Page 6, lines 147-148.
Page 7, lines 149-150. 
It is recommended that the authors adjust for BMI when they assess whether variants of mirRNA-126 and PIK3R1 are associated with T2D and metabolic traits.

Authors

Thank you very much for the recommendation. The aim of the project was to assess the genetic risk factors only. However, we will consider adjustment for BMI in our future studies. 

Minor comments.

Page 16, line 282.

The authors should replace the description “table 9” with “table 8”.

Authors

Thank you very much. Done

Page 2, line 46.

The expression "important global health concerns" does not take an indefinite article.

Authors

Thank you very much. Replaced with the sentence "DM is one of the global health issues due to its bad socio-economic effects on the patients, their families and public health". Changes are highlighted in green.

Page 17, line 307.
The expression "PI3K SNPs" should be replaced with "PIK3R1 SNPs".

Authors

Thank you very much. Done